DNA barcoding and morphological identification of spiny lobsters in South Korean waters: a new record of Panulirus longipes and Panulirus homarus homarus

Hettiarachchi Sachithra Amarin 1 2 3
Hyeon Ji-Yeon 1 4
Mahardini Angka 1
Kim Hyung-Suk 5
Byun Jun-Hwan 1
Kim Han-Jun 6
Jeong Jong-Gyun 7
Yeo Jung-Kyu 7
Kim Shin-Kwon 8
Kim Se-Jae 4
Heo Youn-Seong 7
Sathyadith Jonathan 1 3
Kang Do-Hyung 1 3
Hur Sung-Pyo hursp@kiost.ac.kr 1 3
1 Jeju Marine Research Center, Korea Institute of Ocean Science and Technology (KIOST) , Jeju , Republic of Korea
2 Department of Fisheries and Aquaculture, Faculty of Fisheries and Marine Sciences & Technology, University of Ruhuna , Matara , Sri Lanka
3 Department of Ocean Science, University of Science and Technology , Daejeon , Republic of Korea
4 Department of Biology, Jeju National University , Jeju , Republic of Korea
5 Department of Kinesiology, Jeju National University , Jeju , Republic of Korea
6 Marine Ecosystem Research Center, Korea Institute of Ocean Science & Technology , Busan , Republic of Korea
7 LED-Marine Biology Convergence Technology Research Center, Pukyong National University , Busan , Republic of Korea
8 Aquaculture Research Division, National Institute of Fisheries Science , Busan , Republic of Korea
Waiho Khor
Electronic publication date: 2022 Jan 10
Publication date: 2022
Volume: 10
Electronic Location ID: e12744
Received 2021 Jul 17; Accepted 2021 Dec 14
Copyright: ©2022 Hettiarachchi et al.
Copyright year: 2022
Copyright holder: Hettiarachchi et al.
License: This is an open access article distributed under the terms of the Creative Commons Attribution License, which permits unrestricted use, distribution, reproduction and adaptation in any medium and for any purpose provided that it is properly attributed. For attribution, the original author(s), title, publication source (PeerJ) and either DOI or URL of the article must be cited.
License URL: https://creativecommons.org/licenses/by/4.0/

Keywords: Spiny lobsters, DNA barcoding, Panulirus japonicus, Panulirus longipes, Panulirus homarus homarus, Panulirus stimpsoni, Morphology, Cytochrome oxidase I, Jeju Island, Phylogenetics

Funding: National Institute of Fisheries Science, South Korea R2021012 The Korea Institute of Ocean Science and Technology PE99922 This research was funded by the National Institute of Fisheries Science, South Korea (R2021012) and Korea Institute of Ocean Science and Technology (PE99922). The funders had no roles in study design, data collection and analysis, decision to publish, or preparation of the manuscript.

==============================
To date, 19 species of spiny lobsters from the genus Panulirus have been discovered, of which only P. japonicus, P. penicilatus, P. stimpsoni, and P. versicolor have been documented in South Korean waters. In this study, we aimed to identify and update the current list of spiny lobster species that inhabit South Korean waters based on the morphological features and the phylogenetic profile of cytochrome oxidase I (COI) of mitochondrial DNA (mtDNA). Spiny lobsters were collected from the southern and eastern coasts of Jeju Island, South Korea. Phylogenetic analyses were performed using neighbor-joining (NJ), maximum likelihood (ML), and Bayesian inference (BI) methods. The ML tree was used to determine the spiny lobster lineages, thereby clustering the 17 specimens collected in this study into clades A, B, C, and D, which were reciprocally monophyletic with P. japonicus, P. homarus homarus, P. longipes, and P. stimpsoni, respectively. These clades were also supported by morphological examinations. Interestingly, morphological variations, including the connected pleural and transverse groove at the third abdominal somite, were observed in four specimens that were genetically confirmed as P. japonicus. This finding is novel within the P. japonicus taxonomical reports. Additionally, this study updates the documentation of spiny lobsters inhabiting South Korean waters as P. longipes and P. homarus homarus were recorded for the first time in this region.

Introduction

Studies on the taxonomical status of spiny lobsters have been conducted throughout the Indian, Pacific, and Atlantic Oceans. Nineteen species from the genus Panulirus have been discovered in these regions, of which seven species have been found within the East China Sea (Holthuis, 1991), including P. japonicus, P. penicilatus, P. stimpsoni, and P. versicolor in South Korean waters (Kim et al., 2009). This genus can be identified based on their transverse ridges with clear-cut connections and decalcified areas on the female sternum. In male lobsters, variations can be observed in the copulatory ornamentation and setation (George, 2005).

In addition to morphological observations, genetic information, such as the mitochondrial cytochrome oxidase subunit I (COI) mitochondrial DNA (mtDNA), is used to identify unknown and novel specimens (Meyer & Paulay, 2005). Marine fauna diversity can be assessed using the COI mtDNA through a technique known as DNA barcoding. DNA barcoding has also been used for conservation purposes, such as phylogeographic analysis, invasive species detection, and forensic studies (Meyer & Paulay, 2005; Bucklin, Steinke & Blanco-Bercial, 2011; Senevirathna & Munasinghe, 2013; Sembiring et al., 2015; Leray & Knowlton, 2016). Phylogenetic studies on spiny lobsters have classified them into four (I–IV) non-formal genetic clades (Ptacek et al., 2001). In addition, phylogeographic analysis based on the P. homarus mtDNA profile has suggested discrimination between the western (parapatric isolation, secondary contact, and introgression) and eastern (active peripheral speciation) populations (Farhadi et al., 2017).

Documenting biodiversity is an essential step that could improve the management and conservation of sustainable natural resources. In South Korea, biodiversity studies have been conspicuously initiated since 2007, following the establishment of the National Institute of Biological Resources (NIBR). Consequently, several new species have been discovered in South Korea, and this number is expected to increase continuously, with a potential for recording up to 60,000 species by 2020 (Biodiversity Division, Nature Conservation Bureau& Ministry of Environment, 2014). In the case of marine fishes, Kim (2009) reported that approximately five described species are documented for the first time within the Korean Peninsula every year. Meanwhile, in 2018, 76 species of epibenthic invertebrates were documented within the southern part of the East Sea, Korea and ∼61% of these were identified as decapods (Park & Huh, 2018).

As a part of the general biodiversity documentation, this study aimed to update the existing records by identifying spiny lobster species inhabiting South Korean waters (Jeju Island). The identification was conducted by using molecular (DNA barcoding) and morphological examination. The spiny lobsters used in this study were collected from the southern part of Jeju Island, South Korea, where several branches of Kuroshio currents (Yellow Sea and Tsushima warm currents) are reported to drift. Based on the newly recorded P. homarus homarus and P. longipes as well as the new morphotype of P. japonicus discovered in this study, the list of spiny lobster species and the general species diversity record within the South Korean waters was updated. Thus, the findings of this study will facilitate appropriate regulation and management of spiny lobsters in this area.

Materials & Methods

Sample collection

This study was conducted under the approval of the animal care and use committee of Jeju National University (2020-0012). Adult spiny lobsters (Panulirus spp.) were collected from three sampling sites along the southern coast of Jeju Island, South Korea. Site 1 was located around Hwasun Harbor (33°13′57.5′N 126°19′55.8′E), site 2 was around Seogwipo Harbor (33°13′57.0′N 126°33′53.2′E), and site 3 was around Pyoseon Port (33°19′37.2′N 126°50′49.9′E) (Fig. 1). Spiny lobsters were located by scuba diving at night (8:00 to 11:00 p.m.) from August to November 2020 and caught using hand nets. The collected animals were placed in iceboxes and transported to the Jeju Tropical Seawater Research Facility at the Jeju Marine Research Center of the Korea Institute of Ocean Science and Technology, Jeju Island, South Korea. All lobsters were reared in acrylonitrile butadiene styrene tanks (20 tons) with a constant circulation of seawater (23 ± 1 °C) and fed daily at 16:00 with commercially formulated powder feed (Heukja, Kopec Ltd., Jeonla-Namdo, South Korea) until analysis.

Figure 1 Map showing the sampling sites of spiny lobsters collected by scuba diving along the southern coast line of the Jeju Island.

Examination of morphological features

A total of 17 spiny lobsters were collected for morphological examination. The specimens were anesthetized on ice for 10 min prior to the examination. Body length (BL) and weight (BW) of each lobster were measured, and the detailed features of the body and appendages were photographed for examination (Fig. 2). Morphological features and color markings were examined with reference to the morphological characterization of lobsters described by George & Holthuis (1965). Species identification was done based on the examination of body color, presence of cross bands on antennal and antennular flagella, size and number of spines on the antennular plate, availability of transverse grooves on abdominal segments, existence of stridulating organs, and presence of exopods in the second and third maxillipeds.

Figure 2 Photographs of dorsal and ventral sides of spiny lobster specimens collected from Jeju Island, South Korea.

(A) P. japonicus, (B) P. homarus homarus, (C) P. longipes, and (D) P. stimpsoni.

DNA Extraction and PCR amplification

Genomic DNA was extracted from the muscle of the pereiopod of all 17 spiny lobsters. DNA was extracted using the AccuPrep® Genomic DNA Extraction Kit (Bioneer, Daejeon, South Korea), following the manufacturer’s instructions. Concentration and purity of the extracted DNA were measured using a Thermo Scientific™ NanoDrop™ One microvolume UV-Vis spectrophotometer (Thermo Scientific, Wilmington, DE, USA).

A polymerase chain reaction (PCR) was performed to amplify the mitochondrial COI gene region using the HCO1490/LCO2198 universal primers, specially designed for invertebrates (Folmer et al., 1994). PCR was performed using a 50 µL reaction mixture, consisting of 100 ng genomic DNA, 0.25 µL Taq polymerase (Takara Bio Inc., Shiga, Japan), 5 µL 10X Ex. Taq DNA polymerase buffer (Takara Bio Inc.), 1 µL each of 10 µM forward and reverse primers, and 4 µL (2.5 mM) dNTPs (Takara Bio Inc.). The PCR thermal profile comprised an initial step of 5 min at 94 °C, followed by 30 cycles at 94 °C for 30 s, 50 °C for 30 s, and 72 °C for 45 s, followed by a final extension at 72 °C for 5 min. The amplified PCR products were separated using 1% agarose gel electrophoresis, and target bands were purified using the AccuPrep® PCR Purification Kit (Bioneer), according to the manufacturer’s instructions.

COI cloning and sequencing

Purified PCR products were ligated to the T-easy vector (Takara Bio Inc.). The ligation mixture was prepared with 2 µL ligation buffer, 3 µL T-easy vector, 1 µL T4 ligase, and 4 µL PCR water. Recombinant plasmids were transformed into Escherichia coli JM109 (DE3) (Promega, USA), cultured on Luria-Bertani agar plates supplemented with ampicillin (LB amp+) and incubated overnight at 37 °C. Subsequently, LB broth (4 mL) was inoculated with the grown colonies and incubated overnight at 37 °C, following which plasmids were extracted using the AccuPrep® plasmid extraction kit (Bioneer). Extracted plasmids were sent for sequencing at Macrogen Pvt. Ltd. (South Korea).

Alignment and phylogenetic analyses

COI sequences were edited and aligned with the reference sequences of various spiny lobster species, retrieved from the BOLD system (http://barcodinglife.org/) and the NCBI database (https://www.ncbi.nlm.nih.gov/) using MEGA 7.0 (Kumar et al., 2016). Sequence alignment was generated using a high-throughput MUSCLE method (Edgar, 2004), and subsequent phylogenetic analyses were performed using neighbor-joining (NJ), maximum likelihood (ML), and Bayesian inference (BI) methods. NJ and ML were constructed using MEGA 7.0 (Kumar et al., 2016) and RAxML v8.2.X (Stamatakis, 2014), respectively. Both the analyses were run with 1000 bootstrap replications. In addition to ML, a priori test was performed using jModelTest 0.1.1(Posada, 2008) to determine the best evolutionary model fitted to the current sequences, and the general time-reversible gamma distribution rate parameter (GTR+G) was used to construct the tree (Guindon & Gascuel, 2003; Posada, 2008). Finally, BI was performed for 5,000,000 generations using MrBayes 3.2.7a Ronquist et al. (2012).

Additionally, pairwise mean distances between groups were calculated using MEGA 7.0 to obtain the genetic divergence information. Furthermore, DnaSP v5 (Librado & Rozas, 2009) was used to measure the nucleotide diversity and the number of polymorphic sites within each clade to which the newly recorded Jeju Island spiny lobsters belonged.

Results

Phylogenetic analyses of spiny lobster COI genes

The COI sequences ranging from 526–712 bp were successfully sequenced. Sequence editing and trimming resulted in 338 bp, which were used for phylogenetic analyses. The NCBI GenBank accession numbers obtained for the sequences are provided in Table S1. A phylogenetic tree was constructed using NJ, ML, and BI, with the values on the branch indicating the bootstrap proportion of NJ and ML, followed by the posterior probability from BI analysis. Trees constructed from ML were used to visualize the topological lineage of spiny lobsters. According to the ML tree, spiny lobsters collected from Jeju Island were grouped under four different clades: Clade A, Clade B, Clade C, and Clade D, which were monophyletic with Panulirus japonicus, P. longipes, P. stimpsoni, and P. homarus, respectively (Fig. 3).

Figure 3 Maximum likelihood constructed tree of spiny lobster COI mitochondrial DNA.

The branch values indicate bootstrap proportion of neighbor joining and maximum likelihood followed by posterior probability of Bayesian inference analysis, respectively (maximum proportion is equal to 100). Scale bar represents the number of base substitutions per site.

Clade A included eight COI sequences of spiny lobsters from Jeju Island, which were closely related to P. japonicus from Japan and Taiwan. Of these, seven sequences were shown to be exclusively claded with P. japonicus specimens collected from Japan, with a high bootstrap proportion and posterior probability (NJ/ML/BI; 99/72/92) (Fig. 3). Meanwhile, only one sequence was claded with the P. japonicus specimens collected from Taiwan (NJ/ML/BI; 99/77/99). Intraspecific diversity revealed that seven haplotypes were found exclusively from Jeju Island with Pi value 5.0% ± 2.0% and 59 polymorphic sites (Table 1).

Two COI sequences were included under Clade B. As shown in Fig. 3, this clade was highly supported (NJ/ML/BI; 100/89/100) to be monophyletic with P. longipes as well as its subtypes, the P. longipes longipes and P. longipes fermorstriga. Specifically, the two spiny lobster sequences obtained in this study seemed to share a common ancestor, P. longipes from India (NJ/ML/BI; 67/67/100). However, the intraspecific diversity analysis indicated that both sequences differed slightly from each other, as indicated by the two haplotypes that were identified. The nucleotide diversity (Pi ± SD) was 2.4% ± 1.2% and five polymorphic sites were observed (Table 1).

Among the COI sequences of spiny lobsters collected in the current study, six sequences were clustered into clade C, which included P. stimpsoni from regions such as South China Sea and Hongkong. This clade was supported by 100/99 of NJ/ML bootstrap proportion and 100 posterior probability based on BI analysis (Fig. 3). In this study, four haplotypes of P. stimpsoni were found within Jeju Island. In addition, nucleotide diversity (Pi ± SD) and number of polymorphic sites were reported at 2.4% ± 1.2% and eight, respectively.

Table 1 Intraspecific diversity of spiny lobsters inhabiting Jeju waters.

Clade	Closest taxa	N	Pairwise mean distance between clades (%)	No. of haplotype	Pi ± SD%	No. of polymorphic sites	
			Clade A	Clade B	Clade C	Clade D				
A	Panulirus japonicus	8	0				7	5.0 ± 2.0%	59	
B	Panulirus longipes	2	27.6	0			2	0.6 ± 0.2%	5	
C	Panulirus stimpsoni	6	41.7	34.2	0		4	2.4 ± 1.2%	8	
D	Panulirus homarus homarus	1	33.5	38.0	25.3	0	na	na	na	
Notes.

na data not available due to insufficient sample number

In the current study, only one specimen was claded with the P. homarus group (Clade D). The sequence of this specimen was closely related to the P. homarus homarus from Marquesas Island, French Polynesia, and the topology was strongly supported by bootstrap proportions and posterior probability (NJ/ML/BI; 100/100/100). These two sequences were distinct from the P. homarus collected from Indonesia, Oman, India, Sri Lanka, Iran, and Mozambique. As for the Jeju intraspecific variation, the analysis could not be performed due to the insufficient number of samples.

The pairwise mean distances between clades were run using Tamura 3-parameter in MEGA 7.0. The closest genetic distance was between clade C (P. stimpsoni) and clade D (P. homarus), with 25.3% differences. Meanwhile, the furthest distance was between clade A (P. japonicus) and clade C (P. stimpsoni), with 41.7% differences (Table 1).

Morphological examination

Material examined: P. japonicus (Jeju morphotype); Female; Total Length (TL): 28.5 cm; Body Weight (BW): 551 g; Site 03, Pyoseon Port, Seogwipo, Jeju Island, South Korea; 33°19′37.2′N, 126°50′49.9′E; August 21, 2020; ∼10 m depth.

Description: The carapace is reddish brown in color, and abdominal segments are greenish brown (Fig. 2A). White spots present in the lateral margin of the carapace and the lateral region of the abdomen. Spines of various sizes are randomly scattered on the carapace, and majority of the spine bases are black in color. The mid-dorsal surface of the carapace bears reddish-brown hairs (Fig. 4A). The dorsal surface of the frontal horns is dark greenish-brown and white spots are present. The ventral side of the frontal horn is orange. Frontal margin of the antennular plate armed with two separate medium-sized spines. The inner dorsal side of the antennal peduncles is pinkish in color. The antennules are without cross bands. The ventrolateral margin of the carapace made a reddish-brown soft surface line with white blotches (Fig. 4A). Non-interrupted transverse grooves are visible on the dorsal surface of each abdominal segment (Fig. 5A). Posteriorly directed hair is present in both the posterior margins of the somites and transverse grooves. Transverse grooves are curved at the lateral end of the second, third, and fourth somites, and interconnected with the corresponding pleural grooves in the first and third somites (Figs. 6A, 6B). The transverse groove in the second somite ends up too close to its pleural groove, making a pseudo connection. Both second and third maxillipeds bear exopods (Figs. 7A, 7B, 7C).

Figure 4 Photographs of lateral side of the carapace of spiny lobsters collected from Jeju Island, South Korea.

Arrows with the lowercase letters of each photograph indicate the following morphological features respectively; (A) P. japonicus, fh: Dark greenish brown frontal horns with white spots and orange color ventral margin; s: Randomly scattered spines on carapace with black color basal area; lM: White spotted lateral margin of the carapace; Enlarged area demarcated by rectangle: Reddish brown hairs on the mid-dorsal area of the carapace. (B) P. homarus homarus, fh: Frontal horns covered with white spots which create pseudo cross-bands; lg: Blue color lateral groove; aps: Conspicuous spines on antennular plate; s: Spines with white color basal area in lateral region of the carapace; Enlarged area demarcated by rectangle: Brownish yellow stiff hairs around the base of the spinules on posterior-lateral area of the carapace. (C) P. longipes, fh: Brown color frontal horns with orange color tips; ms: Conspicuous two medium size spines stand behind the frontal horns; bs: Black color spine; s: Spine with orange color tip and white color base; Enlarged area demarcated by rectangle: Reddish brown hairs on posterior mid dorsal area of the carapace. (D) P. stimpsoni, fh: Reddish brown frontal horns with four pale yellow cross-bands; ss: Two small size conspicuous spines behind the frontal horns; s: Brown tipped small spine on carapace; lM: White lateral margin of the carapace; wl: White line over the lateral margin of the carapace; Enlarged area demarcated by rectangle: Pale yellow color hairs on the mid-dorsal area of the carapace. fh: frontal horns; s:spine; lm: lateral margin; aps: antennular plate spine; lg: lateral groove; ms: medium size spine; ss: small size spine; wl: white line.

Figure 5 Photographs of dorsal side of the abdomen somites of spiny lobsters collected from Jeju Island, south Korea.

Arrows with the lowercase letters of each photograph indicate the following morphological features respectively; (A) P. japonicus a: Non-interrupted transverse groove with posteriorly directed hairs; b: Posterior margin of the second somite with posteriorly directed hairs; c: Brown to purplish color base of telson; d: Reddish brown and slightly curved posterior margin of telson. (B) P. homarus homarus a: Shallow scallops in anterior margin of the transverse groove; b: Posterior margin of the second somite; c: Numerous white spots in posterior somites; d: Greenish brown and rounded posterior margin of telson. (C) P. longipes a: Non-interrupted transverse groove with posteriorly directed hairs; b: Posterior margin of the second somite with hairs; c: Median notched of transverse grooves in second, third and fourth somites; d: Numerous white and orange color spots in distal part of the abdomen; e: Dark brown line followed by white line in posterior margin of the telson. (D) P. stimpsoni a: Stiff hairs on dorsal surface of the abdomen somite; b: Disturbed pattern of hair and slight depression in mid-dorsal region of the second, third and fourth somites; c: Reddish brown and pale yellow color spots in posterior part of the abdomen; d: Reddish brown line followed by white line at the rounded posterior margin of the telson.

Figure 6 Photographs of lateral side of the abdominal somites of spiny lobsters collected from Jeju Island, South Korea and their diagrammatic view.

Arrows with the lowercase letters of each photograph indicate the following morphological features respectively; (A, B) P. japonicus (A) Posteriorly directed hairs in transverse groove; (B) posteriorly directed hairs in posterior margin of the second somite; (C) Curved transverse groove at the lateral end of second,third and fourth somites; (D) Interconnection between transverse groove and pleural groove at the first somite; (E) Interconnection between transverse groove and pleural groove at the third somite; (F) Transverse groove is ending up very close to pleural groove of the second somite. (C, D) P. homarus homarus a: Shallow scallops on anterior margin of the transverse groove; b: Crenulated articulation at lateral end of the transverse groove; c: White spot in anterior-lateral region of the fourth somite (present in each somite except the first); d: Numerous white spots on posterior somites. (E, F) P. longipes (A) Short stiff hairs on transverse groove in the first somite; (B) Posteriorly directed hairs in transverse groove; (C) Posteriorly directed hairs in posterior margin of the somite; (D) Tubercles in anterior margin of the pleuron of somite II; (E) White spot in the anterior-lateral region of the fourth somite (present in each somite); (F) Conspicuous white spot on pleuron (present in all pleura except the first). (G, H) P. stimpsoni a: Slight depressions and associated firm hairs on dorsal side of the second , third and fourth somites; b: White spot in anteriolateral region of the fourth somite (present in each somite except the first); (C) Four conspicuous teeth (descending order in size) in posterior margin of the pleuron with hairs.

Material examined: P. japonicus (Holthuis morphotype); Female; Total Legnth (TL): 31 cm; Body Weight (BW): 1023 g; site 03, Pyoseon Port, Seogwipo, Jeju Island, South Korea; 33°19′37.2′N, 126°50′49.9′E; August 21, 2020; ∼10 m depth.

Figure 7 Photographs and diagrammatic views of the distinguished mouth parts of four spiny lobster species.

(A) Photograph of mouth region, (B) second maxilliped, (C) third maxilliped of P. japonicus; (D) Photograph of mouth region, (E) second maxilliped, (F) third maxilliped of P. homarus homarus; (G) Photograph of mouth region, (H) second maxilliped, (I) third maxilliped of P. longipes; (J) Photograph of mouth region, (K) second maxilliped, (L) third maxilliped of P. stimpsoni. b, basis; c, carpus; d, dactyl; ex, exopod; fm, first maxilliped; i, ishium; m, merus; p, propodus; sm, second maxilliped; tm, third maxilliped. Scale bars represent 1 cm.

Description: Base color of the carapace is reddish brown, and abdominal segments are greenish brown. Lateral margin of the carapace and lateral region of the abdomen contain white spots. Spines which are varying in sizes with black bases are scattered on the carapace. The mid-dorsal surface of the carapace bears reddish-brown hairs (Fig. S1). Frontal horns are dark greenish brown and white spots are present. Two separate medium size spines are present at the frontal margin of the antennular plate. The antennules are without cross bands. Reddish brown soft surface line margin the ventrolateral region of the carapace. White blotches are present on that line. Each abdominal segment contains non-interrupted transverse groove with posteriorly directed hairs on its posterior margin. Transverse grooves are curved at the lateral region of the second, third and fourth somites. There are no interconnections of transverse grooves with corresponding pleural grooves in the second and third somites (Fig. S2). Both second and third maxillipeds bear exopods.

Material examined: P. homarus homarus ; Female; TL: 37 cm; BW: 1390 g; Site 01, Hwasun Harbor, Seogwipo, Jeju Island, South Korea; 33°13′57.5′N, 126°19′55.8′E; August 18, 2020; ∼10 m depth.

Description: The carapace and front part of the abdominal region are brownish to greenish in color. Antennal and antennular peduncles, walking legs, posterior part of the abdominal region, peduncles of the uropod, and base of the telson are greenish in color (Fig. 2B). The body consists of numerous white spots, which are apparent in the posterior part of the abdomen. Both antennular peduncles and antennular flagella contain white cross bands. The distal part of each antennular peduncle contains white blotches. There are four well-separated spines on the antennular plate and randomly scattered small spines between them. The horns are covered with white spots, creating pseudo bands, and the frontal horn tips are orange (Fig. 4B). The post cervical groove, lateral grooves, and the anterior margin of the carapace are blue. Furthermore, the posterior margin of the antennular plate consists of a triangular design with blue and orange colors. Additionally, non-interrupted transverse grooves are present in each abdominal segment, and the anterior margin of the grooves form shallow scallops (Fig. 5B). The transverse grooves form a crenulated articulation at the lateral area of every somite (Figs. 6C, 6D). A white spot is present in the anterolateral region of each somite. The second maxilliped bears an exopod, while it is absent in the third one (Figs. 7D, 7E, 7F).

Material examined: P. longipes ; Female; TL: 28 cm; BW: 667 g; Site 01, Hwasun Harbor, Seogwipo, Jeju Island, South Korea; 33°13′57.5′N, 126°19′55.8′E; August 18, 2020; ∼15 m depth

Description: The body and the post-orbital area are brown, while the dorsal parts of the carapace and anterior segments of the abdominal region are dark brown in color (Fig. 2C). Both carapace and abdominal regions are covered with randomly scattered white and orange spots. The base of the antennal peduncle is purple, and its dorsal inner side is pinkish brown. The frontal margin of the antennular plate is armed with two separate small spines. Frontal horns are brown, while the tips are orange, and there are two conspicuous medium-sized spines behind the frontal horns. The carapace bears numerous spines of different sizes; some are black, while some have orange tips with white bases (Fig. 4C). Antennular flagella are cross-banded in white. Legs bear orange and white spots at the end of each segment. Orange lines on the lateral and ventral legs are broken into irregular blotches. Transverse grooves are present in every abdominal segment and connect laterally to the pleural grooves (Fig. 5C, Figs. 6E, 6F). Exopods are present in the second and third maxillipeds (Figs. 7G, 7H, 7I).

Material examined: P. stimpsoni ; Female; TL: 28 cm; BW: 765 g; Site 02, Seogwipo Harbor, Seogwipo, Jeju Island, South Korea; 33°13′57.0′N, 126°33′53.2′E; April 06, 2020; ∼15 m depth

Description: Body color is brown to olive green. The posterior part of the cervical groove on the carapace is darker in color (Fig. 2D). Frontal horns are reddish brown in color, tips are pale yellow, and four cross lines of the same color are present. The lateral margin of the carapace forms a white line, and another white line starting from the anterior part of the cervical groove extends over it (Fig. 4D). The lateral region between these two lines is light orangish yellow. The antennule plate is armed with four well-separated spines projected among the hairs, and the frontal pair of spines is larger than that at the posterior region. Antennules are reddish brown with white cross bands. The distal part of each antennule peduncle shows white blotches. Pereiopods are reddish brown with longitudinal white lines and blotches in the propodus, carpus, and merus. The abdominal segments are olive green in color. Reddish brown and pale-yellow speckles are present in each segment. Transverse grooves are not present in any of the abdominal segments. Instead, slight depressions associated with firm hairs are present in the middle of the second, third, and fourth somites. These grooves were disturbed in the middle and became broad on the lateral sides (Fig. 5D). A white spot was present at the top-notch of the triangular plate in the first somite. The same conspicuous white spots are present in the anterolateral region of each somite, except in the second somite, where it is present as a posteriorly interrupted white line (Figs. 6G, 6F). The posterior margin of each pleuron bears four conspicuous teeth associated with hairs, arranged in descending order of size. The second maxilliped bears an exopod, but the third maxilliped does not (Figs. 7J, 7K, 7L).

The major comparative morphological features used for the identification of spiny lobsters are shown in Table 2.

Table 2 The comparative morphological identification features of spiny lobster species collected from Jeju Island and other two species which were previously reported from South Korea.

Feature	P. japonicus (Jeju type)	P. japonicus (George & Holthuis, 1965)	P. homarus homarus	P. longipes	P. stimpsoni	P. penicilliatus	P. versicolor	
No. of examined specimens	4	4	1	2	6	na	na	
Color of carapace	Reddish brown	Reddish brown	Brownish to greenish	Dark brown with white and orange color spots	Brownish to olive green	Reddish brown with many pale yellow spots	Brownish to greenish and black blotches defined with white lines	
Cross bands in antennular flagella	No cross bands	No cross bands	Cross banded	Cross banded	Cross banded	No cross bands	No cross bands	
Spines on antennular plate	Separated medium size 02 spines	Separated medium size 02 spines	Separated 04 spines	Separated 04 spines	Separated 04 spines, frontal pair is larger than hind	04 equal size spines fused at the base	Separated 2 pairs of spines which is unequal in size	
Nature of the ventrolateral margin of the carapace	Reddish brown soft surfaced line with white blotches	Reddish brown soft surfaced line with white blotches	Pale white line edging with numerous soft hairs	Reddish dark brown line	White line and another white line over it	Reddish brown soft surfaced line with white blotches	Pale white line with black line over it	
Color of frontal horns	Dorsal surface is dark greenish brown with white spots, Ventral side is orange color	Dorsal surface is dark greenish brown with white spots, Ventral side is orange color	Tips are orange, pseudo bands made by white spots	brown in color, tips are orange	reddish brown with pale yellow four cross lines and tips	Dark brown dorsal surface with pale white tips	Dark brownish to black with white lines	
Spines behind the frontal horns	02 medium size spines with 03 conspicuous small spines in-between	02 medium size spines with 03 conspicuous small spines in-between	02 medium size spines, no small spines in-between	02 medium size spines with 04 conspicuous small spines in-between	02 medium size spines, no small spines in-between	02 medium size spines	02 medium size spines, no small spines in-between	
Spines in the frontal edge of the epistome	03 spines with spinules in-between	03 spines with spinules in-between	03 spines, no spinules in -between	03 main spines and many spinules in-between	03 spines, no spinules in-between, middle is larger than side spines	03 spines with spinules in-between	na	
Color of abdominal segments	Greenish brown	Greenish brown	Olive green	Dark brown with pale white spots	Greenish	Greenish brown	Brownish blue with white lines along the posterior margin of each somite	
Presence of transverse grooves on abdominal segments	Non-interrupted transverse grooves	Non-interrupted transverse grooves	Non-interrupted transverse grooves and anterior margins are crenulated	Non-interrupted transverse grooves with a middle notch on 2nd, 3rd and 4th segments	No transverse grooves	Non-interrupted transverse grooves	No transverse grooves	
Connection between transverse grooves and pleural grooves	They are not connected in second somite (but very close to the pleural groove)	They are not connected in second and third somites (distinct space between pleural and transverse grooves)	1st transverse groove connected to the pleural groove with vertical shape and others connected the with different articulations like scalloped margins	Transvers groves are connected to the pleural grove in each somite	Not applicable	Not connected	Not applicable	
Presence of exopod in 2nd maxilliped	Present	Present	Present	Present	Present	Present	Present	
Presence of exopod in and 3rd maxilliped	Present	Present	Absent	Present	Absent	Absent	Absent	

Discussion

This study was conducted to update the records of spiny lobster species in South Korean waters, based on the specimen collection in Jeju Island. A previous study identified four spiny lobster species from the genus Panulirus sporadically distributed in South Korean waters: P. japonicus, P. penicilatus, P. stimpsoni, and P. versicolor. However, only P. stimpsoni has been reported in Jeju Island waters (Kim et al., 2009).

In the current study, spiny lobsters were identified using DNA barcoding and morphological analysis. A maximum likelihood tree of COI marker was constructed by including the spiny lobster sequences available in the Genbank and the BOLD system. The results confirmed that the spiny lobsters collected in this study belong to clade A, B, C, and D, which are reciprocally monophyletic with P. japonicus, P. longipes, P. stimpsoni, and P. homarus, respectively. Among the four clades, the closest genetic distance was observed between intra Jeju Island specific spiny lobsters from clade C (P. stimpsoni) and D (P. homarus). On the other hand, the furthest distance was found between clade A (P. japonicus) and C (P. stimpsoni). These results provided additional evidence supporting the previous studies which state that spiny lobsters (Genus: Panulirus) can be morphologically and phylogenetically diversified into two major lineages (Ptacek et al., 2001; George, 2005).

Intraspecific Jeju diversity analysis revealed that clade A appears to have the highest divergence rate based on the number of haplotypes, nucleotide diversity (Pi) and polymorphic sites. Notably, among the collected specimens, majority of the sequences were observed in this clade and thus, were genetically identified as P. japonicus. Among these sequences, one sequence was exclusively sub-claded with P. japonicus from Taiwan, while the remaining were sub-claded with those from Japan. Therefore, it can be assumed that the P. japonicus specimens found in Jeju Island originated from the same population source as those from Japan and Taiwan. Our findings were consistent with those of previous studies, which state that P. japonicus across populations within the Japan, Taiwan, and southern Chinese waters have originated from the same larval pool that mixed within the Kuroshio counter-current region (Inoue et al., 2007; Chan, Yang & Wakabayashi, 2019). Furthermore, this research partially supported the previous hypothesis regarding non-existence of sub-divisions in the P. japonicus population within its spatial distribution (Chan, Yang & Wakabayashi, 2019).

Despite the monophyletic lineages, two morphotypes of P. japonicus were found in the current study, which will be addressed as original type (George & Holthuis, 1965) and Jeju type. Four of the eight P. japonicus specimens found in the current study appeared to be Jeju type based on the variations observed in its morphological features compared to the original type (George & Holthuis, 1965; Kim et al., 2009). The Jeju type specimens had a connected pleural and transverse groove at the third abdominal somite, unlike the holotype from Japan (George & Holthuis, 1965). Moreover, the transverse groove ends close to the pleural groove in the second abdominal somite, forming a pseudo connection, which creates a noticeable gap in the previously described specimens. Morphological variations in spiny lobsters have been discovered in the species P. homarus and P. longipes (Sekiguchi, 1991; Lavery et al., 2014). In addition to this, the evolutionary divergence and phenotypic adaptation that results from geographical dispersal were shown in the different colorations and abdominal patterns among P. homarus morphotypes observed in this study. The phylogenetic clusters clearly supported these apparent variations based on population; thus, each was introduced as a different subspecies (Lavery et al., 2014). In the case of P. japonicus from Jeju Island, each morphotype does not indicate a distinct genetic cluster or an intra spatial subdivision. It is likely that the Jeju type variation occurred as part of an adaptive response due to selective forces and/or environmental restrictions during developmental and settlement stages (George, 2005; Vieira et al., 2016).

This study confirmed that the Jeju Island spiny lobsters in clade B were P. longipes, based on morphological and phylogenetic species concepts (George & Holthuis, 1965; Ravago & Juinio, 2002). The intraspecific diversity of mtDNA COI revealed that, unlike its sister clade (clade A), clade B had a relatively low divergence based on its Pi value and polymorphic sites. In the genus Panulirus, P. longipes is one of the two spiny lobsters that have subspecies because of its morphotype variations (George & Holthuis, 1965). Around the East China Sea region, the subspecies P. longipes longipes and P. longipes femoristriga are distributed throughout southwestern Japan, including Okinawa and Yaeyama Island, through the northern side of Taiwan (Sekiguchi, 1991). The discovery of P. longipes in Jeju Island has updated the records of spiny lobsters inhabiting South Korean waters in general.

In clade C, six spiny lobsters had a well-supported monophyletic relation with P. stimpsoni from Hong Kong and the South China Sea. Compared to the other spiny lobsters in the current study, the phylogenetic analysis of P. stimpsoni remains understudied. This might partially be due to the limited distribution of this species within the East and the South China Sea (Holthuis, 1991; Liu & Cui, 2011). However, a previous phylogenetic study based on genomic sequences of this species confirmed that the lineage of P. stimpsoni among the Decapoda is monophyletic with P. japonicus under the Palinura clade, which is closest to the infraorder Astacidea (Liu & Cui, 2011).

In agreement with the COI result, morphological characterization of the specimen from clade D confirmed that it was identical to the P. stimpsoni holotype from Hong Kong (Holthuis, 1991). The presence of this species in South Korean waters has also been reported in the southern part of Jeju Island (Kim et al., 2009). Morphologically, some features were remarkably similar to those of P. versicolor. However, a feature in P. versicolor, such as a distinct continuous white transverse band along the edge of its first to sixth abdominal somite posteriors, is the key to differentiate between these two species (Holthuis, 1991; Kim et al., 2009).

As reported previously, the current study also showed that a divergent monophyletic pattern could be observed within the subspecies of P. homarus (Lavery et al., 2014; Singh et al., 2017). A spiny lobster found in Jeju Island waters appeared to be closely related to the subspecies P. homarus homarus in Marquesas Island, French Polynesia (Ptacek et al., 2001). This finding is worth further discussion with adequate number of samples. Although the morphological features of Jeju Island P. homarus were identical to those described by George & Holthuis (1965), in concordance to its COI sequence, this specimen was further confirmed to belong to the Homarus morphotypes based on its dark green coloration and the presence of microsculpta (Berry, 1974; Holthuis, 1991). This corroboration of P. homarus has updated the list of spiny lobster species inhabiting the South Korean coastal region.

Conclusions

This study aimed to identify the spiny lobsters found in Jeju Island by using a phylogenetic and morphological approach. Based on the results of this study, it can be concluded that the use of DNA barcoding technique along with morphological examination could enhance clarity regarding the evolutionary and taxonomical position of spiny lobsters from Jeju Island among those from different population regions. In this regard, two of the four spiny lobster species identified in this study, P. longipes and P. homarus homarus were recorded for the first time in South Korean waters. In addition, morphological variations discovered in the Jeju type P. japonicus seem to be the first record within P. japonicus taxonomical studies. As the record has been updated, the results of this study could aid in updating the global conservation status of spiny lobsters. However, further phylogeography and population genetic studies are necessary for better understanding of their evolutionary and diversity status. This study implied that considerable biodiversity remains undiscovered in South Korea, and DNA barcoding could be a potential tool for unraveling the biodiversity.

Supplemental Information

Supplemental Information 1 Photographs of dorsal and ventral sides, lateral side of the carapace and dorsal side of the abdomen somites of P. japonicus (Holthuis morphotype) collected from Jeju Island, South Korea

A) Dorsal side, B) Ventral side; C) Lateral side of the carapace; fh: Dark greenish brown frontal horns with white spots and orange color ventral margin; s: Randomly scattered spines on carapace with black color basal area; lm: White spotted lateral margin of the carapace; Enlarged area demarcated by rectangle: Reddish brown hairs on the mid-dorsal area of the carapace. D) Dorsal side of the abdomeinal somites; a: Non-interrupted transverse groove with posteriorly directed hairs; b: Posterior margin of the second somite with posteriorly directed hairs; c: Brown to purplish color base of telson; d: Reddish brown and slightly curved posterior margin of telson.

Click here for additional data file.

Supplemental Information 2 Photograph of lateral side of the abdominal somites of P. japonicus (Holthuis morphotype) collected from Jeju Island, South Korea and it’s diagrammatic view

Arrows with a lowercase letter of photograph are indicating the following morphological features respectively; a: Posteriorly directed hairs in transverse groove; b: Posteriorly directed hairs in posterior margin of the second somite; c: Curved transverse groove at the lateral end of second, third and fourth somites; d: Interconnection between transverse groove and pleural groove at first somite; e: Gap between transverse groove and pleural groove at third somite; f: Gap between transverse groove and pleural groove at second somite. Scale bar represents two cm.

Click here for additional data file.

Supplemental Information 3 NCBI GenBank accession numbers obtained for COI sequences and sampling sites of spiny lobsters collected for this study

Click here for additional data file.

The authors are thankful to Dr. Chulhong Oh, Ms. Moonjeong Lee and Dr. Soojin Heo of the Korea Institute of Ocean Science and Technology (KIOST) for their expert assistance and helpful suggestions.

Additional Information and Declarations

Competing Interests

Author Contributions

Field Study Permissions

Data Availability

The authors declare there are no competing interests.

Sachithra Amarin Hettiarachchi conceived and designed the experiments, performed the experiments, analyzed the data, prepared figures and/or tables, authored or reviewed drafts of the paper, all the drawings were done, and approved the final draft.

Ji-Yeon Hyeon and Angka Mahardini conceived and designed the experiments, performed the experiments, analyzed the data, prepared figures and/or tables, authored or reviewed drafts of the paper, and approved the final draft.

Hyung-Suk Kim and Han-Jun Kim performed the experiments, authored or reviewed drafts of the paper, collected spiny lobsters by Scuba diving, and approved the final draft.

Jun-Hwan Byun performed the experiments, analyzed the data, prepared figures and/or tables, authored or reviewed drafts of the paper, and approved the final draft.

Jong-Gyun Jeong, Jung-Kyu Yeo and Shin-Kwon Kim performed the experiments, authored or reviewed drafts of the paper, and approved the final draft.

Se-Jae Kim, Youn-Seong Heo and Do-Hyung Kang conceived and designed the experiments, authored or reviewed drafts of the paper, and approved the final draft.

Jonathan Sathyadith analyzed the data, authored or reviewed drafts of the paper, and approved the final draft.

Sung-Pyo Hur conceived and designed the experiments, performed the experiments, analyzed the data, prepared figures and/or tables, authored or reviewed drafts of the paper, collected spiny lobsters by Scuba diving, and approved the final draft.

The following information was supplied relating to field study approvals (i.e., approving body and any reference numbers):

The Animal care and use committee of the Jeju National University approved the study (2020-0012).

The following information was supplied regarding data availability:

The COI gene sequences are available at GenBank: MZ203547 to MZ203550 and OK037046 to OK037058.

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
