# Peer review of "DNA barcoding and morphological identification of spiny lobsters in South Korean waters: a new record of Panulirus longipes and Panulirus homarus homarus"

_PeerJ, doi:10.7717/peerj.12744_

## Round 0.1 · original submission · Major Revisions

I would agree with the reviewers that although this manuscript reports on the new record of two Panulirus species, it is severely limited by only one representative specimen for each species. It would definitely enhance the results of the manuscript if the authors could include more specimens into the results. I highly doubt that only one specimen of each species was acquired from a 4-month sampling period? However, if this is really the case, the authors should discuss this as well (the sampling frequency is not described in the manuscript). The use of only one sample would render many analyses, especially morphological comparison to be inconclusive, as pointed out by reviewers as well.

Specific comments:
1. Ensure that all 'Panulirus' are in italic.
2. Please add, in Figure 1, which site each analysed specimen was from.
3. The Introduction and Discussion sections should be re-written, following reviewers' comments.

Reviewer 1 ·

Basic reporting

The authors have data set, although I noted lack of details on the number of samples used. In addition, the manuscript is clearly written in a professional manner.

The English language could be improved. It would be better if proof reading done by a colleague who is proficient in English and familiar with the subject matter review your manuscript, or contact a professional editing service.

Experimental design

Research question well defined.

Do you have field study permits?

The methodology for DNA work used in this study had been written clearly. Missing info: How many samples had been sequenced per species? Multiple samples sequence data could be anayzed to determine intraspecific genetic diversity (for population at Jeju Isalnd).

Similarly, referring to morphological examination, there is no statement saying how many samples had been examined per species. I could argue that the variations in morphology -differences between Jeju sample with George and Holthuis (1965) is just a coincidence (if only one sample had been examined).

Validity of the findings

Morphological examination findings had been presented well, with photographs and labels. DNA data had been analysed according to standard procedure. The only information need is the number of samples examined in this study, per species.

Additional comments

Abstract : good
Introduction : good
Materials & methods -Indicate how many samples had been used in DNA work and morphological work.

Results : Line 205, you missed a dot after ‘spines’?

Discussion: Good discussion. Minor corrections needed for example you can use word ‘reciprocally monophyletic’ , 'phylogenetic species concept', etc. in explaining the phylogenetic tree topology. Again, if you only examined one sample per species, discussion section should be written with caution.
You named each clade with genus' name, which then confusing whether you should used italic and capital each word or otherwise. Perhaps use clade A, clade B, etc., instead of clade Japonicus, etc. to solve this confusion.

Conclusion
Conclusion should be short and sweet, highlighting your best result/achievement and how it impact scientific knowledge /management /general public.
“Based on morphological data and COI gene sequence analysis, four species namely…….. with first record of ??? in Jeju Island. COI gene information is useful to aid species id? This info will help in….Future study need to involve more samples??? In order to get …..”

Reviewer 2 ·

Basic reporting

This is a straightforward study describing two new records of spiny lobster from the genus Panulirus based on molecular and morphological examination. The experimental is mostly acceptable given the simplicity of the study, though not fully competent. The major limitation is the scope of the study is small, with only a few sites and one specimen from each of the four species. Therefore, though I am convinced that P. longpipes and P h. homarus represent new record to Korean water, I am not convinced by most of the discussions concerning the genetic divergence or morphological variations for the studied specimens compared to the data in GenBank or descriptions in literature. I will leave it to the editor’s call for whether two new records can justify a paper in PeerJ, of which should fit boarder readers’ interest, but I do think substantial revisions shall be made in the introduction and discussion to focus more on the solid finding whether than inconclusive descriptions on genetic divergence. The authors should focus on the recent finding on biodiversity on S Korean water and/or potential for range expansion of more tropical species to Korea (either natural or anthropogenic).

Experimental design

Simple study, so no major fault in methodology except sample size is small.

Validity of the findings

As aforementioned, the two records should be valid, but those discussions on genetic divergence, morphological variations are inconclusive given the small sample size or irrelevant to major context. For example, “drifting of P. japonicus” individual from Taiwan to Korea” is apparently misleading and erroneous because specimens of P. japonicus split into two clades in the tree by the authors (one with Jeju, another the other with Japan). One of this may represent cryptic species (to be confirmed). This may partially explain the morphological variation from the holotype (which is from Japan).

It looks strange to me that “PL was genetically confirmed as Panulirus longipes”; “The PS specimen was clustered into the Stimpsoni clade with P. stimpsoni”. The abbreviation is by species ID so apparently a circular reasoning here.

Table 2: descriptions for the other two species found in Korean water (penicilatus and versicolor that were not sampled) shall be included for comparison to ease the readers.

Additional comments

Introduction includes a large portion of information about Panulirus phylogeny or population genetics that is not relevant to the current study, whilst not much background for recent discovery or biodiversity in Korean S water is mentioned.

Reviewer 3 ·

Basic reporting

The manuscript is written with some grammatical errors. English language should be improved. Please change and just use Panulirus homarus (PH) or P. homarus in the manuscript because I found many mistakes regarding that scientific name. The literature reference and data provided are sufficient.

Experimental design

The research experimental design is sufficient and reliable. This research is very important as there is very limited data on the marine lobster in the database.

Validity of the findings

The manuscript reported the finding of the new species of lobsters in Korean waters which are important for the future studies of marine lobsters. The findings are reliable due to the identification based on morphological and molecular approaches.

Additional comments

Due to many species of Panulirus sp. with diverse morphological characteristics. if possible, I just want to suggest the authors classify back the species of these four lobsters (PH, PS, PJ, and PL) based on other references together with George and Holthuis (1965) to avoid misidentification.

---

## Round 0.2 · Minor Revisions

Thank you authors for your revision. The current version is very much improved. However, Reviewer 2 has some minor comments that I think are quite helpful in improving the manuscript. Please address them accordingly and hope to receive the revised copy soon.

Reviewer 1 ·

Basic reporting

The revise version is clear and unambiguous. The authors had used English professional service. This manuscript also contains adequate literature references. The article structure is good with clear figures and tables.

Experimental design

This manuscript is an original primary research, within the aim and scope of the journal. It had been successful in filling the identified knowledge gap. It follows the requirement of science technical and ethical standard. Methods are described with sufficient details.

Validity of the findings

The findings are valid, useful for broad audience because of its biodiversity theme (and conservation). Good data with robust analysis and good discussion. Conclusions are well-written.

Additional comments

I am happy with the improvement made by the authors. Well done!

Reviewer 2 ·

Basic reporting

The authors have done a good job in revising the manuscript and included more sequences to strengthen the finding. i think the manuscript is acceptable, and i only have two additional comments on discussion for the authors to consider:

1/ line 327 -333: this is apparently unnecessary as the study mostly re-analyzed previously published sequence data for most lobster species, so it is not surprising that the inferred phylogeny is more or less the same. Line 329-333 is clearly too descriptive and can be deleted.

2/ line 390-412: the discussion here is pre-mature, Lavery et al 2014 already discussed this issue in detail with much boarder sampling. The authors only included one new P. homorus sequence but do a complicated (and somehow confusing) discussion that i think will only have negative impact on the manuscript quality.

Experimental design

It is a reasonably well performed study.

Validity of the findings

The finding is valid, though some discussions are irrelevant as stated above.

---

## Round 0.3 · accepted · Accept

Thank you all authors for following through all the concerns and suggestions from the reviewers. The current version of the manuscript has improved significantly and is acceptable for publication in PeerJ.